# Hypercalcemia Following Adrenalectomy for Cushing Syndrome in a Patient with Post-Surgical Hypoparathyroidism

**DOI:** 10.3390/diseases13010020

**Published:** 2025-01-17

**Authors:** Pietro Locantore, Alessandro Oliva, Gianluca Cera, Rosa Maria Paragliola, Roberto Novizio, Caterina Policola, Andrea Corsello, Alfredo Pontecorvi

**Affiliations:** 1Unit of Endocrinology, Department of Translational Medicine and Surgery, Università Cattolica del Sacro Cuore, Fondazione Policlinico “A. Gemelli” IRCCS, Largo Gemelli 8, 00168 Rome, Italy; pietro.locantore@icloud.com (P.L.);; 2Unicamillus, Saint Camillus International University of Medical Sciences, Via di S. Alessandro 10, 00131 Rome, Italy; 3Division of Endocrine Surgery, Ospedale Isola Tiberina-Gemelli Isola, 00186 Rome, Italy

**Keywords:** hypercalcemia, hypoparathyroidism, Cushing syndrome, adrenalectomy

## Abstract

**Background:** Hypercalcemia is a frequently encountered laboratory finding in endocrinology, warranting accurate clinical and laboratory evaluation to identify its cause. While primary hyperparathyroidism and malignancies represent the most common causes, many other etiologies have been described, including some reports of hypercalcemia secondary to adrenal insufficiency. On the contrary, hypoparathyroidism is a relatively common cause of hypocalcemia, often arising as a complication of thyroid surgery. In real-world clinical practice, however, many challenges come into play, and a comprehensive approach may not be enough to establish a diagnosis. **Case presentation:** we describe a peculiar case of severe hypercalcemia occurring in a 47-year-old woman with a previous history of post-surgical permanent hypoparathyroidism treated with calcitriol (0.5 µg bid) and calcium carbonate (1 g qd), which persisted after withdrawal of these drugs. During her follow-up, an ACTH-independent Cushing syndrome was diagnosed, leading to a unilateral right adrenalectomy. In the two months following surgery, she was admitted to the emergency ward on three occasions because of severe, persistent, idiopathic hypercalcemia. On each occasion, parathyroid hormone levels were confirmed to be undetectable, with low vitamin D levels. Common and rare causes of hypercalcemia were excluded, and the persistence of severely elevated calcium levels led to the empirical use of intravenous clodronate, achieving remission of both hypercalcemia and, unexpectedly, hypoparathyroidism. After 8 months, due to borderline-reduced calcium, calcitriol at 0.5 µg qd was restarted. After 18 months of follow-up, the patient is well and normocalcemic, with low-dose calcitriol. Notably, the patient had no acute adrenal insufficiency, distinguishing this case from other post-adrenalectomy hypercalcemia reports. **Conclusions:** the history of hypoparathyroidism makes this case even more unusual, and it encourages careful follow-up of hypoparathyroid patients with Cushing syndrome. Ongoing observation, as well as new research on the physiopathology of cortisol and calcium metabolism, are needed to clarify the pathogenesis of this case.

## 1. Introduction

Hypercalcemia is an uncommon laboratory finding in clinical practice, and it may be a sentinel for serious health issues, hence the need to properly assess its etiology. Hyperparathyroidism accounts for the vast majority of causes of hypercalcemia (~54%), followed by malignancies (~36%). Several other less frequent causes exist, including calcium-alkali and milk-alkali syndrome, granulomatous diseases (e.g., sarcoidosis, tuberculosis, etc.), Glucocorticoid Withdrawal Syndrome (GWS), vitamin A excess, immobilization, and hyperthyroidism [1,2,3]. Moreover, several medications can cause or contribute to hypercalcemia (Table 1).

Hypercalcemia can be classified as PTH-dependent or PTH-independent [2]. In fact, in the presence of hypercalcemia, high levels (or inappropriately normal levels) of serum PTH are usually sufficient to diagnose hyperparathyroidism. Confirmation of its origin requires the assessment of complete calcium–phosphorus metabolism, including serum and urinary calcium and phosphate, PTH and 25-OH vitamin D levels, renal function, and imaging. Instead, hypercalcemia in malignancies is usually associated with low serum PTH. It may be caused by several mechanisms, such as the production of PTH-related protein (PTHrp), bone osteolysis (i.e., metastasis and myeloma), or excessive extra-renal activation of vitamin D into calcitriol, as in lymphomas [4] and other malignancies [5]. More rarely, a tumor may produce PTH ectopically, as described mostly in lung cancer [6]. When these two main causes are ruled out, rarer causes of hypercalcemia need to be investigated.

PTHrp physiologically increases during pregnancy and lactation, potentially leading to hypercalcemia during pregnancy, after delivery, or during lactation [7,8,9].

Calcium-alkali syndrome is a novel clinical entity that has almost completely taken the place of milk-alkali syndrome, caused by inappropriate intake of calcium supplements [10].

Some drugs have been shown to cause hypercalcemia, with some widely used, such as thiazides, and others being more ‘unusual’, such as voriconazole and itraconazole, foscarnet, growth hormone, or even omeprazole [11]. Lithium therapy is a cause of PTH-dependent hypercalcemia because of the induced secretion of PTH and subsequent bone resorption. Additionally, a ‘rebound’ hypercalcemia effect has been described after discontinuation of denosumab or teriparatide therapy. Hypercalcemia has also been infrequently reported with the use of sodium–glucose transport inhibitors (SGLT2i), immune checkpoint inhibitors (ICIs), theophylline, aromatase inhibitors, and abaloparatide [2].

Granulomatous diseases are a large family of conditions that are characterized by the presence of granulomas generated by macrophages in chronic inflammatory conditions. Macrophages are one of the extra-renal sources of 1,25(OH)_2_D_3_ [12], and thus granulomatous diseases are often characterized by hypercalcemia [13,14]. Clinical presentation, elevated levels of 1,25(OH)_2_D_3_, and high markers of inflammation can guide the clinician in the correct diagnosis.

Patients who are immobilized for a prolonged period, especially in the elderly and those suffering from high bone remodeling (Paget disease, osteolytic metastases), have a higher risk of developing hypercalcemia [15].

Lastly, rare reports of hypercalcemia of unclear origin have been reported in patients with various conditions, including rhabdomyolysis, acromegaly, pheochromocytoma, kidney tubular acidosis, epilepsy following a ketogenic diet, and COVID-19, and in healthy people after hours of intense exercise [2]. This complication has been documented in patients with rheumatoid arthritis [16], pericarditis caused by Nocardia [17], and major burns complicated by renal failure and immobilization [18]. Vitamin A excess has also been described as a rare cause of hypercalcemia, attributed to increased bone resorption [19], and also in the setting of burn treatment [20]. Additionally, hyperthyroidism represents another condition associated with hypercalcemia due to enhanced bone resorption [21].

Hypercalcemia can also be caused by a rapid decrease in glucocorticoid levels in the context of so-called ‘Glucocorticoid Withdrawal Syndrome’ (GWS), which can occur after adrenalectomy, after discontinuation of corticosteroid therapy, or in Addison’s Disease [22,23]. The mechanisms causing hypercalcemia in this setting are still unclear. Hypercortisolism inhibits osteoblast and osteocyte function, mildly increases bone resorption, reduces intestinal calcium absorption, and increases renal calcium excretion, leading to hypercalciuria and negative calcium balance [24,25]. After adrenalectomy for ACTH-independent Cushing syndrome, bone turnover declines and fracture risk normalizes [26,27].

On the other hand, hypoparathyroidism is a well-known cause of hypocalcemia, representing a rare but fundamental complication of thyroid surgery. It is characterized by transient or permanent hypocalcemia, often needing calcitriol and/or calcium supplementation [28]. In most cases, serum calcium levels normalize within a few weeks or months after surgery. In around 4–0.9% of patients undergoing thyroidectomy, instead, hypoparathyroidism persists after 1 year and is considered permanent, even if some patients have been reported to recover up to 8–16 years after thyroid surgery [28,29,30].

Here, we present a peculiar case of persistent hypercalcemia occurring after adrenalectomy in a patient with permanent hypoparathyroidism.

Hypercalcemia in post-surgical hypoparathyroid patients may seem contradictory. Some authors have reported this scenario, usually due to calcium-alkali syndrome because of calcium and/or vitamin D derivatives excess [31,32,33,34,35,36] or the onset of another concomitant disease [37]. These cases, however, show a recurrence of hypocalcemia when the cause of hypercalcemia ceases (i.e., discontinuation of supplements or treatment of the underlying condition). Our patient, instead, showed persistent hypercalcemia, suggesting a more complex clinical picture.

## 2. Case Presentation

C.P., a 47-year-old woman, came to our attention in March 2022 with suspicion of Cushing syndrome and a right adrenal mass.

Her medical history was relevant for post-surgical hypothyroidism and hypoparathyroidism following total thyroidectomy in 2013 for papillary thyroid cancer (histology: a 6 mm unifocal papillary thyroid cancer, classic variant in the left lobe without extrathyroidal extension or vascular invasion; pT1N0 R0). She was treated with thyroxine 650 mcg/week, calcitriol 0.75 μg qd, and calcium carbonate 1000 mg qd. During her follow-up, she had experienced several episodes of tetany with hypocalcemia, which recovered with calcium infusions.

She had no evidence of recurrent or persistent thyroid cancer. The patient also had a history of hyperhomocysteinemia, uterine fibroids, and an ovarian cyst.

No family member had relevant diseases. She had a daughter in good health, delivered through cesarean section in 2005.

Her recent history started a few months before, with an increase in body weight and the onset of progressively worsening hyperglycemia, hypertension, and hyperlipidemia. Given the uncontrolled glycemic profile, therapy with multiple daily injections of insulin was started. In November 2021, she suffered from persistent lumbar pain caused by a L4-L5 disc herniation and was later admitted to another hospital for acute kidney injury following the use of NSAIDs as painkillers. While hospitalized, after recovery of kidney function, an iliofemoral and inferior vena cava thrombosis with pulmonary embolism was diagnosed. On that occasion, a chest CT incidentally showed the presence of a right adrenal mass. Since that time, she also presented secondary amenorrhea.

Given the clinical picture of hyperglycemia, hypertension, venous thromboembolism, amenorrhea, increasing body weight with central obesity, and facial plethora, the finding of an adrenal mass eventually raised the suspicion of Cushing syndrome. She underwent an adrenal MRI that confirmed the presence of the adrenal nodule (max. diameter: 30 mm). Therefore, she was referred to our Endocrinology Unit in March 2022.

Upon examination, the patient showed some cushingoid features: facial plethora, central obesity, proximal muscle weakness, and hypertension. No striae rubrae nor buffalo hump were observed.

At the time, she was taking levothyroxine 650 μg/week, calcitriol 0.5 μg h 8:00 + 0.25 μg h 20:00, calcium carbonate 1000 mg qd, pantoprazole 20 mg qd, apixaban 5 mg bid, potassium chloride 600 mg qd, folic acid supplement, perindopril 10 mg qd, indapamide 2.5 mg qd, amlodipine 10 mg qd, rosuvastatin 20 mg qd, ezetimibe 10 mg qd, metformin 500 mg tid, semaglutide 0.50 mg qwk, and basal and prandial insulin therapy.

Laboratory testing was performed, confirming the diagnosis of ACTH-independent Cushing syndrome based on the high levels of 24 h urinary free cortisol (UFC), late-night salivary and serum cortisol, and insuppressible cortisol after an overnight suppression test (dexamethasone 1 mg). Her thyroid and parathyroid disease were also reassessed (Table 2).

In order to characterize the adrenal mass, the patient underwent a new abdomen MRI, which did not observe the typical loss of signal of adrenal adenomas, leaning toward a diagnosis of lipid-poor adenoma.

For that, the patient was referred to the Endocrine Surgery Unit. Ketoconazole 200 mg bid was started as a bridge medical therapy while waiting for surgery, with good control of the disease (UFC 196 nmol/24 h).

A presurgical chest–abdomen–pelvis CT (June 2022) confirmed the presence of the 3 cm lipid-poor adenoma based on the wash-out features of the adrenal mass (absolute wash-out: 31%; relative wash-out: 22%). The pulmonary embolism appeared to be resolved, but a residual parietal inferior vena cava thrombosis was still apparent. As collateral findings, the CT showed severe steatosis with multiple hepatic angiomas, a splenic microcyst, and uterine fibroids. No relevant alterations were found in any of the remaining organs.

In July 2022, she underwent a right adrenalectomy through a laparoscopic lateral transperitoneal approach. After surgery, the patient started replacement therapy with intravenous hydrocortisone 100 mg tid in the first 24 h after surgery, gradually tapered in the following days, and shifted to oral cortisone acetate at 75 mg per day (50 mg + 25 mg). The histopathology report confirmed the diagnosis of an adrenal cortical adenoma with oncocytic features.

Serum creatinine and electrolytes were stable one week after discharge. Blood pressure measurements improved after surgery, and antihypertensive therapy was gradually discontinued. Cortisone acetate and insulin therapy were progressively reduced. Due to occasional tachycardia, a beta-blocker (bisoprolol 2.5 mg) was started. A laboratory and clinical follow-up was scheduled for September 2022.

In August 2022, one month after surgery, the patient was admitted to the emergency ward of another hospital because of marked asthenia, malaise, nausea, and vomiting. Laboratory testing showed hypercalcemia (3.39 mmol/L), and she was treated with an intravenous saline infusion. Calcium carbonate and calcitriol were discontinued. At discharge, calcium levels were lower (2.30 mmol/L) and, because of her post-surgical hypoparathyroidism, calcitriol (0.5 + 0.25 μg qd) was reintroduced.

Two weeks after discharge (September 2022), these symptoms recurred. Her blood tests showed a recurrence of severe hypercalcemia (3.89 mmol/L, 15.6 mg/dL), and she was once again admitted to the emergency ward of our hospital. The patient denied taking calcium or any supplement or changing her dietary habits, including dairy products. She confirmed that she consumed the correct prescribed dose of calcitriol. Calcitriol was temporarily interrupted, and the possibility of new causes of hypercalcemia was investigated (Table 3).

Given the increase in serum creatinine values, apixaban was replaced with enoxaparin, and a nephrologist was consulted. The worsening renal function, however, seemed to be an effect, rather than a possible cause, of hypercalcemia. After hydration, kidney function recovered. PTH remained undetectable. At that moment, the patient was also taking cortisone acetate 25 mg qd. There was no clinical or laboratory evidence of adrenal insufficiency. In the acute phase, hydrocortisone was increased intravenously and then progressively reduced. The patient remained hyperglycemic, in good control with basal insulin therapy.

Unfortunately, laboratory evaluation of TRACP-5b, bone alkaline phosphatase, and PTHrp levels was unavailable at the time of hospitalization. However, other markers of bone metabolisms were evaluated, and imaging was performed to exclude occult neoplastic disease. A bilateral mammography and a whole-body CT showed no evidence of adrenal disease (Figure 1) or neoplastic or granulomatous disease. A ^99m^Tc-HMDP bone scan was negative for bone lesions.

After excluding the main causes of hypercalcemia, a diagnosis of calcitriol intoxication was hypothesized. However, considering the normal values of 1,25(OH)_2_-vitamin D (120 pmol/L, 50 pg/mL), normal kidney function before and after the acute episode, the corrected calcium value of 2.2 mmol/L after appropriate intravenous hydration, and undetectable parathyroid hormone values, calcitriol was reintroduced at a lower dose of 0.25 μg bid. The patient was informed of the need for careful adherence to the treatment, with no neglect or abuse, and of a regular follow-up plan. She was then discharged.

During tight biochemical follow-up, plasmatic calcium was normal–high (2.59 mmol/L), and calcitriol was fully discontinued again. A careful calcium follow-up was started due to the history of severe hypocalcemia and recent episodes of hypercalcemia.

Nevertheless, 3 weeks after the previous admission (October 2022), hypercalcemia recurred once again (3.87 mmol/L), requiring a new hospitalization due to the new presence of the patient’s usual symptoms (Table 3 and Figure 2).

A ^18^F-FDG PET-CT scan was negative for occult disease. An abdominal ultrasound revealed bilateral renal microlithiasis.

A psychiatric consultation was conducted to evaluate the possibility of calcium/vitamin D supplement abuse, which the patient denied once more, which was confirmed by her relatives. The consultant identified anxious traits with paroxysmal exacerbations. No clues of possible calcium-alkali syndrome were found.

Considering the persistent and worsening hypercalcemia, this time unresponsive to intravenous hydration and diuretic and corticosteroid therapy, empiric therapy with intravenous clodronate 300 mg was administered over two subsequent days. A short-acting bisphosphonate was preferred [38] due to the history of hypoparathyroidism and the possible risk of subsequent hypocalcemia and the occurrence of a possible ‘hungry bone’ syndrome [39]. In the following days, calcium levels gradually decreased down to a minimum of 1.8 mmol/L (corrected for hypoalbuminemia: 1.975 mmol/L). Appropriate supplementation of calcium carbonate 1 g intravenously qd and alpha-calcidiol was started until the stabilization of calcium levels. A less active vitamin D supplement was preferred. The patient was discharged with calcium carbonate 2 grams bid + alpha-calcidiol 0.25 μg qd. Calcium carbonate and alpha-calcidiol were gradually interrupted, and the patient has been stable and normocalcemic with no therapy for 8 months. After 8 months, however, calcium returned to borderline-reduced levels, and calcitriol 0.5 μg qd was reintroduced. Since then, the patient has been stable and normocalcemic up to a follow-up of 18 months.

## 3. Discussion

The above-described case of unexplained hypercalcemia represents an intriguing diagnostic challenge, where many factors may be involved in altering bone and calcium homeostasis, including the prior history of hypoparathyroidism treated with calcium and calcitriol, Cushing syndrome, the subsequent adrenalectomy, and the acute kidney injury.

As cited, the most common cause of hypercalcemia is primary hyperparathyroidism. In our patient, undetectable serum PTH and several episodes of tetany confirmed the patient’s history of iatrogenic hypoparathyroidism. Therefore, a possible recovery of parathyroid activity can be excluded, and a PTH-independent cause of hypercalcemia had to be investigated.

No signs suggestive of recurrent or new malignancy were observed. Total body CT, ^18^F-FDG PET-CT scan, and bone scintigraphy ruled out bone involvement or possible causes of PTHrp secretion. The deteriorating renal function detected upon admission appeared to be an effect of severe hypercalcemia, as it fully recovered in the intercritical phases and the patient did not show any sign of dehydration. Thyroid function tests were normal on thyroxine treatment. No indication of errors in medication intake was found. In summary, our clinical assessment, history evaluation, laboratory exams, and imaging techniques, were able to exclude the great majority of the causes of hypercalcemia listed above (Figure 3). The most likely hypotheses that were considered are discussed below.

The first hypothesis was a possible calcium-alkali syndrome. This is in accordance with the already cited rare reports of hypercalcemia in hypoparathyroid patients due to calcium or calcitriol excess. Given the history of calcium and calcitriol supplementation, we could not completely rule out the possibility of inappropriate calcium supplementation. Excess calcium intake may determine hypercalcemia and reduced renal function with normal or low plasmatic phosphorus levels, all of which were present in our patient. However, calcium-alkali syndrome is usually cured by calcium intake regulation, saline infusion, and patient education [40]. This was not the case in our patient, as hypercalcemia recurred despite the discontinuation of calcium supplementation following the first hospitalization and the reduction of calcitriol intake to 0.25 μg qd.

Some patients with hypoparathyroidism may also show calcium-alkali syndrome caused by calcium and/or excessive supplementation of calcitriol [34], alphacalcidol [32,35], and other long-acting active vitamin D derivatives, such as dihydrotachysterol [33,36]. In our patient, calcitriol was completely discontinued during the second hospitalization, without any decrease in plasmatic calcium levels until the tenth day from admission, after the use of bisphosphonates. Additionally, blood gas analyses were performed and showed a trend towards acidosis rather than alkalosis. Lastly, the psychiatric evaluation did not find any clue of potential self-harm and/or Munchausen syndrome that could explain occult calcium excess intake.

A second possibility was that of a variant of GWS. Patients with hypercortisolism who undergo unilateral adrenalectomy almost invariably suffer from transient adrenal insufficiency, as the prolonged cortisol excess inhibits the hypothalamus–pituitary–adrenal (HPA) axis. Immediately after surgery, the HPA axis remains suppressed, with low ACTH and cortisol levels, requiring glucocorticoid supplementation. During the following weeks or months, ACTH levels start to rise, so glucocorticoid supplementation may be progressively reduced. However, several months and up to 5 years are needed to regain adequate adrenal function [41]. Different surgical techniques have been associated with various rates of perioperative complications and length of hospitalization, but with varying results, suggesting a personalized approach for each patient [42]. These potentially different outcomes, however, do not have an impact on HPA axis suppression. Therefore, peri- and post-operative glucocorticoid replacement therapy is necessary to avoid adrenal crises [43]. Our patient indeed showed low cortisol and inappropriately low ACTH levels. Adrenal insufficiency was confirmed during her third hospitalization with an ACTH stimulation test (basal cortisol: 147.3 nmol/L or 53.4 ng/mL; peak cortisol: 322.4 nmol/L or 116.8 ng/mL; delta: 63.4 ng/mL), so glucocorticoid replacement therapy was continued. For this, we hypothesized that our patient experienced a variant of a GWS due to the rapid fall of cortisol levels from endogenous hypercortisolism to a replacement therapy, even if adequate.

If glucocorticoid therapy is discontinued or insufficient, patients can develop adrenal insufficiency and GWS with hypercalcemia. Some authors have reported hypercalcemia in patients who underwent adrenal surgery and stopped glucocorticoid supplementation [22]. This scenario is part of the GWS, in which hypercalcemia is linked to adrenal insufficiency. This may occur also in patients treated with glucocorticoids for other reasons if glucocorticoids are abruptly discontinued. Notably, Walker and Davies [23] reported the occurrence of hypercalcemia in a patient with hypoparathyroidism and Addison’s disease. The causes of increased calcium levels in this context, however, are still unclear. Some authors have proposed that it may be explained by hemoconcentration, reduced renal calcium excretion, increased bone resorption, and heightened intestinal calcium absorption [44,45]. In contrast, other authors have found intestinal calcium absorption or the parathyroid hormone to be of no influence on calcium levels in this setting. Additionally, they have hypothesized another mechanism: glucocorticoid deficiency may cause a loss of inhibition of TSH secretion, leading to increased thyroxine levels and subsequent bone resorption [45,46,47]. In all of these cases, appropriate glucocorticoid replacement therapy achieved the normalization of calcium homeostasis.

These explanations, however, seem to be insufficient to explain our case. First, our patient was on cortisone acetate at a sufficient dose (at least 25 mg qd) and it was never discontinued, and no signs or symptoms of adrenal crisis ever occurred.

Nevertheless, we speculated that rather than a complete glucocorticoid withdrawal, she suffered from a relative reduction in glucocorticoids after removing the cortisol-producing adrenal mass. Indeed, during her first hospitalization, we found lower urinary calcium (1.43 mmol/24 h or 57 mg/24 h), which could be explained by a theoretical reduction in the hypercalciuric effect of hypercortisolism. Inconsistent with this hypothesis, however, are our findings during the second hospitalization: we found a normal urinary calcium output and normal osteocalcin and β-cross-laps.

Ultimately, several data point towards a ‘bone-driven’ hypercalcemia. First, the fractional calcium excretion (FECa) was not reduced, excluding renal disorders as a primary direct cause of hypercalcemia. Second, the renal reabsorption of phosphate, indicated by TRP and TmP/GFR, was in the upper-normal range, which is compatible with the known hypoparathyroidism. With the improvement of hypercalcemia, we observed a slight reduction of phosphate, TRP, and TmP/GFR, compatible with the hypothesis of an increased bone turnover. Last, the rapid and effective response to clodronate therapy suggests that hypercalcemia may have been a result of an altered equilibrium in bone metabolism. In fact, increased bone resorption may cause hypercalcemia and may recede after bisphosphonate therapy, but bone resorption markers were not elevated. However, the patient’s post-surgical hypoparathyroidism might have impaired the reliability of these markers. In fact, hypoparathyroidism has been associated with reduced bone turnover and mineralization due to the absence of the effect of PTH on osteoblasts [24]. Cortisol excess, instead, is associated with altered bone metabolism because of its effects on osteoblasts, osteocytes, and osteoclasts [48]. Thus, in our patient, the absence of PTH should have caused low bone turnover markers. Hypercortisolism, instead, may have gradually restored normal levels of these markers.

In conclusion, our opinion is that this case may be considered as a variant of GWS. After surgery, hypercortisolism ceased and normal cortisol levels were restored. This may have caused a GWS-like phenomenon. The osteoclast–osteoblast activity may have been unbalanced, potentially causing hypercalcemia, even in the absence of overt adrenal crisis.

Research is needed to better clarify the pathophysiological processes of bone metabolism in cortisol and parathyroid diseases. Further observation will be needed to shed some light on this case.

## 4. Conclusions

In real-world clinical practice, patients often present with intricate scenarios. In this case, the patient’s unexplained hypercalcemia presented a complex diagnostic challenge that remains unexplained. The empiric treatment with clodronate achieved a successful resolution of hypercalcemia, but this raises questions about the underlying mechanisms. GWS has some common features with our patient’s presentation, but it cannot fully explain this case. This report highlights the importance of closely monitoring patients with hypoparathyroidism and Cushing syndrome, especially when they undergo surgery. Bone pathophysiology can be deeply altered in the case of concomitant parathyroid hormone and cortisol alterations, affecting calcium homeostasis, as demonstrated in this case. The absence of PTH modifies bone structure and function; the action of cortisol, or at least of hypercortisolism, on such an altered bone may cause unpredictable scenarios. Future research will shed light on such a complex interplay.

## Figures and Tables

**Figure 1 diseases-13-00020-f001:**
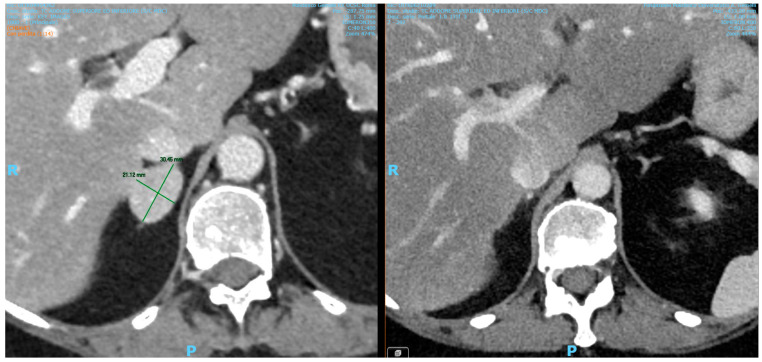
Pre- and post-surgical CT scan images of the right adrenal adenoma showing complete macroscopic resection.

**Figure 2 diseases-13-00020-f002:**
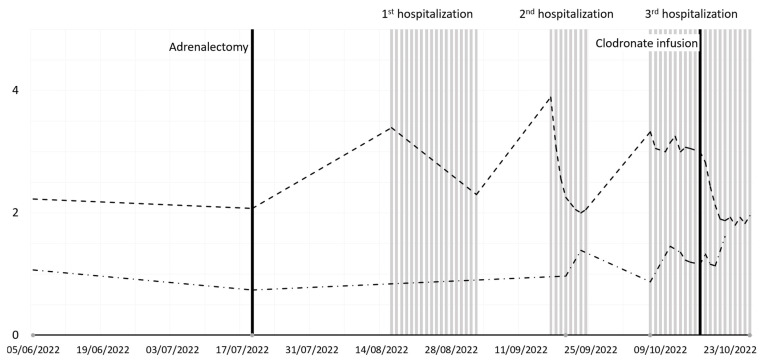
Serum calcium, phosphate, and PTH levels during follow-up.

**Figure 3 diseases-13-00020-f003:**
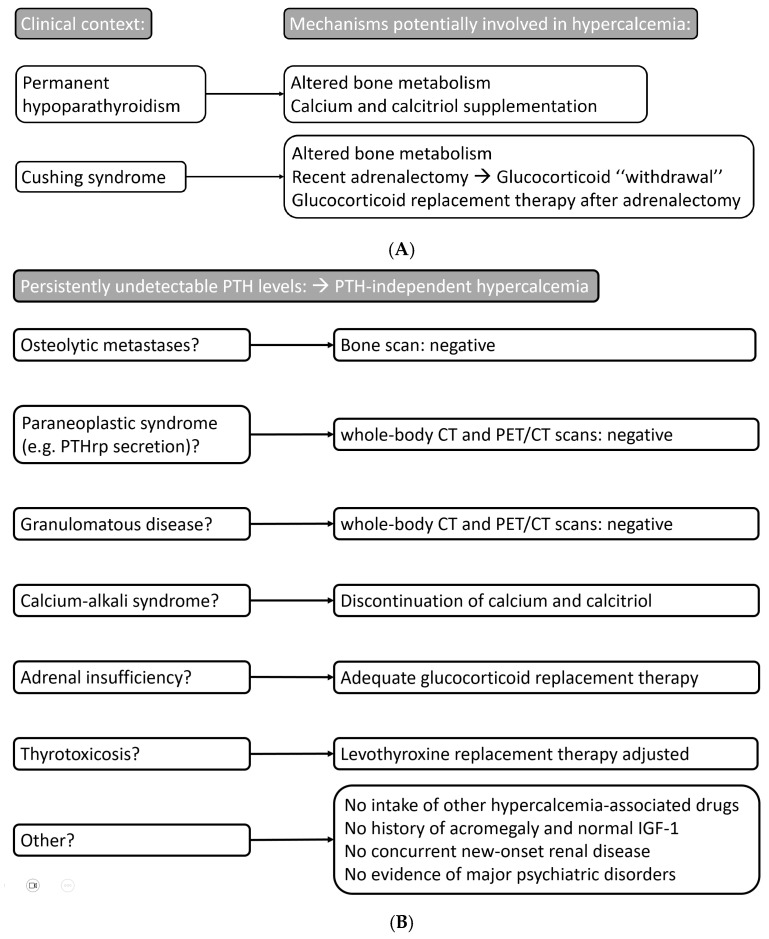
Summary of diagnostic and therapeutic considerations. (**A**) Pathophysiological background of the case report. (**B**) Causes of hypercalcemia and diagnostic–therapeutic considerations in this case.

**Table 1 diseases-13-00020-t001:** Main causes of hypercalcemia. GWS: Glucocorticoid Withdrawal Syndrome.

PTH-dependent causes: Primary (and tertiary) hyperparathyroidism Familial hypocalciuric hypercalcemia
Malignancies (e.g., osteolytic bone metastases)
PTHrp ectopic secretion
Drug-induced (calcitriol, calcium, thiazides, lithium, vitamin A)
Granulomatous diseases (increased 1,25(OH)_2_-vitamin D levels)
Immobilization
Others (hyperthyroidism, acromegaly, GWS, Bartter syndrome)

**Table 2 diseases-13-00020-t002:** Laboratory exams, March 2022. Cushing’s syndrome diagnosis. UFC: urinary free cortisol. Normal reference ranges are in italics.

Laboratory Exam	March 2022
ACTH (2.22–12.1 pmol/L)	<1.1 pmol/L
Serum cortisol (h 8:00) (166–607 nmol/L)	745 nmol/L
Serum cortisol (h 24:00)	684 nmol/L
Night salivary cortisol (<3.03 nmol/L)	23.7 nmol/L (0.86 μg/dL)
1 mg dexamethasone overnight suppression test (h 8:00 cortisol) (<50 nmol/L)	665 nmol/L (241 ng/mL)
UFC (mass spectrometry) (<193 nmol/24 h)	1305 nmol/24 h (473 μg/24 h)
TSH (0.35–3.2 µIU/mL)	0.30 µIU/mL
fT4 (10.9–21.2 pmol/L)	18.7 pmol/L
Thyroglobulin (Tg)	0.4 ng/mL
Anti-Tg antibodies (<4.5 IU/mL)	<1.3 IU/mL
Parathyroid hormone (PTH) (1.5–7.6 pmol/L)	<0.2 pmol/L
Calcium (2.17–2.57 mmol/L)	2.17 mmol/L
Phosphate (0.80–1.61 mmol/L)	1.23 mmol/L
Serum glucose (3.61–5.55 mmol/L)	12.37 mmol/L (223 mg/dL)
HbA1C (23–41 mmol/mol)	62.0 mmol/mol
Serum creatinine (44.2–88.4 µmol/L)	76.0 µmol/L (0.86 mg/dL)
Sodium (135–145 mmol/L)	139 mmol/L
Potassium (3.5–5.0 mmol/L)	3.3 mmol/L

**Table 3 diseases-13-00020-t003:** Laboratory exams.

Laboratory Exam	September 2022	October 2022
ACTH (2.22–12.1 pmol/L)	2.42	22.9
Serum cortisol (166–607 nmol/L)	55	455
ACTH-stimulation test (250 μg)		
Serum cortisol (basal) (nmol/L)		147.3
Serum cortisol (peak) (nmol/L)		322
TSH (0.35–3.2 µIU/mL)	1.59	1.42
fT4 (10.9–21.2 pmol/L)	15.1	18.3
Thyroglobulin (Tg) (ng/mL)	0.7	1.1
Anti-Tg antibodies (<4.5 IU/mL)	<1.3	<1.3
Parathyroid hormone (PTH) (1.5–7.6 pmol/L)	<0.2	<0.2
25OH-vitamin D (77.4–249.5 nmol/L)		62.1
1,25(OH)_2_-vitamin D (58–206 pmol/L)	120	72
Albumin (34–48 g/L)	32	34
Calcium (at admission) (2.17–2.57 mmol/L)	3.08	3.88
Calcium (at discharge) (2.17–2.57 mmol/L)	2.05	1.95
Phosphate (0.80–1.61 mmol/L)	1.39	0.87
Magnesium (0.66–1.07 mmol/L)	0.52	0.71
Alkaline phosphatase (0.77–1.93 µkat/L)	1.42	1.65
Osteocalcin (10.0–45.0 ng/mL)		33.5
Beta-cross-laps (0.2–1.0 ng/mL)		0.4
Serum glucose (3.61–5.55 mmol/L)	6.99	5.88
HbA1c (23–41 mmol/mol)	59	
Hb (120–150 g/L)	109	138
Serum creatinine (at admission) (44.2–88.4 µmol/L)	180	158
eGFR (≥90 mL/min/1.73 m^2^)	28	33
Serum creatinine (at discharge) (44.2–88.4 µmol/L)	121	93
eGFR (≥90 mL/min/1.73 m^2^)	45	63
Serum HCO_3_^−^ (24–28 mmol/L)		24
Serum electrophoresis		No alterations
Sodium (135–145 mmol/L)	141	136
Potassium (3.5–5.0 mmol/L)	3.4	3.1
Urinary calcium (3.5–7.5 mmol/24 h)	1.43	9.65
Urinary phosphate (7.3–58.0 mmol/24 h)	4.32	19.8
Urinary creatinine (≤15.9 mmol/24 h)	9.1	9.3
FECa (>0.01)	0.01	0.051
%TRP (85–95%)	93.7	62.9
Tmp/GFR (0.84–1.23 mmol/L)	1.3	0.55
Proteinuria (≤0.229 g/24 h)		1.8

## Data Availability

The data presented in this study are available upon request from the corresponding author due to privacy, legal, and ethical reasons.

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
