# Peer review of "Hypercalcemia Following Adrenalectomy for Cushing Syndrome in a Patient with Post-Surgical Hypoparathyroidism"

_diseases, 2025, doi:10.3390/diseases13010020_

Round 1
Reviewer 1 Report
Comments and Suggestions for Authors
diseases-3357191
"Hypercalcemia Following Adrenalectomy for Cushing Syndrome in a Patient with Post-Surgical Hypoparathyroidism" by Locantore, et al.
General comments:
The authors investigated and discussed the possible causes of hypercalcemia secondary to adrenal insufficiency, in which endogenous PTH secretion was not involved. The clinical course and the past history of this patient can provide an interesting insight on the mechanism of "hypocortisolemia-induced hypercalcemia", which we sometimes encountered in the clinical setting. The theme and focus of this report may be interesting, and the findings included novelty; however, several points should be addressed to improve the manuscript.
Specific comments:
1. The introduction is quite long as a case report. The part from "1.2" can be in the discussion part.
2. The case presentation for the clinical course is complicated, and therefore it should be graphed. The graph would be better to contain the key laboratory data such as corrected Ca, iP, cortisol, ACTH, PTH and the medication.
3. To follow up the balance of Ca-iP controlled by non-PTH effects, the authors should calculate %TRP, %FECa and Tmp/GFR levels during the clinical courses. Also, the related bone metabolic data such as TRACP-5b, bone-ALP (BAP), etc. and PTH-related protein (PTHrP) should be shown.
4. To clarify the condition of adrenal Cushing's syndrome, pre- and post-surgical CT scan would be informative in this report.
5. The mechanism of hypercalcemia after Cushing's adrenalectomy may involve the changes of PTH sensitivity. How were the possibility to utilize calcimimetics?
6. The reviewer recommends to put the conceptive graphical summary to conclude the present consideration for the wide-range of the readers.
Reviewer 2 Report
Comments and Suggestions for Authors
This is a case report of unexplained transient hypercalcemia after surgical treatment for adrenal Cushing syndrome in a patient previously known to have postsurgical hypoparathyroidism.
The introduction is unnecessarily long, I suggest massive shortening. The case presentation is clear but the opinion of the authors regarding a potential explanation should be included! Just discussing alternative explanation is not enough. In my opinion for example a milk-alkali syndrome can not be reasonably excluded and I favor this explanation but the opinion of the authors should also be presented.
Reviewer 3 Report
Comments and Suggestions for Authors
Recommendations:
1. I disagree that hypercalcemia is frequent, is usually 1%. And majority are asymptomatic.
2. Please add in the discussion section a mini-review of literature, otherwise the article is rather monotonous.
3. No figures, images related to the case are seen here, especially those who reveal the tumor.
4. Add in the discussion section a brief mentioning of adrenal tumor, see this: https://doi.org/10.3390/diagnostics13213351
5. Reference list more homogenous.
6. Auto-citation for 4 authors.
Round 2
Reviewer 1 Report
Comments and Suggestions for Authors
To the Authors:
General comments:
The authors appropriately revised their manuscript, which clarified discussion regarding the mechanism of "hypocortisolemia-induced hypercalcemia" after the adrenalectomy in a patient with no PTH secretion. There are still some minor concerns to publish this case report to improve the manuscript.
In the Table 3, the changes on data of TRP, FECa and Tmp/GFR before and after the administration of bisphosphonate are also interesting point in this case. The author need to shortly discuss the added data. Possibly, the changes of FECa and TRP are in parallel with serum levels of Ca and iP after the bisphosphonate treatment under the absence of endogenous PTH. In addition, the data of %FECa in the Table may not indicate the unit of “percentage”, please check it.
Author Response
General comments:
The authors appropriately revised their manuscript, which clarified discussion regarding the mechanism of "hypocortisolemia-induced hypercalcemia" after the adrenalectomy in a patient with no PTH secretion. There are still some minor concerns to publish this case report to improve the manuscript.
In the Table 3, the changes on data of TRP, FECa and Tmp/GFR before and after the administration of bisphosphonate are also interesting point in this case. The author need to shortly discuss the added data. Possibly, the changes of FECa and TRP are in parallel with serum levels of Ca and iP after the bisphosphonate treatment under the absence of endogenous PTH. In addition, the data of %FECa in the Table may not indicate the unit of “percentage”, please check it.
Thank you again for your comments, that contribute to a more stimulating discussion on this case.
Indeed, FECa, TRP, and TmP/GFR have provided further insights into renal handling of calcium and phosphate. We have added two sentences commenting these results in the last paragraphs of the discussion (page 10). We have also removed the percentage unit of FECa in the Table, as suggested.
Reviewer 2 Report
Comments and Suggestions for Authors
This is a case report of unexplained transient hypercalcemia after surgical treatment for adrenal Cushing syndrome in a patient previously known to have postsurgical hypoparathyroidism.
The manuscript has been improved and is of reasonable interest to the readers for it to be considered for publication in my opinion.
Normal range for lab tests should be included in the tables!!!
Author Response
This is a case report of unexplained transient hypercalcemia after surgical treatment for adrenal Cushing syndrome in a patient previously known to have postsurgical hypoparathyroidism.
The manuscript has been improved and is of reasonable interest to the readers for it to be considered for publication in my opinion.
Normal range for lab tests should be included in the tables!!!
Thank you very much for your comments.
We have added normal ranges in the tables, as suggested.
Reviewer 3 Report
Comments and Suggestions for Authors
Congratulations to the authors! Good luck!
Author Response
Congratulations to the authors! Good luck!
Thank you for your feedback and support.